# Efficiency of Basil Essential Oil Antimicrobial Agents under Different Shading Treatments and Harvest Times

**Zoran S. Ilić** [1,\*], **Lidija Milenković** [1], **Ljubomir Šunić** [1], **Nadica Tmušić** [1], **Jasna Mastilović** [2], **Žarko Kevrešan** [2], **Ljiljana Stanojević** [3], **Bojana Danilović** [3] and **Jelena Stanojević** [3]

[1] Faculty of Agriculture, University of Priština in Kosovska Mitrovica, 38219 Lešak, Serbia; lidija.milenkovic@pr.ac.rs (L.M.); ljubomir.sunic@pr.ac.rs (L.Š.); nadica.tmusic@pr.ac.rs (N.T.)
[2] Institute of Food Technology, University of Novi Sad, 21000 Novi Sad, Serbia; jasna.mastilovic@fins.uns.ac.rs (J.M.); zarko.kevresan@fins.uns.ac.rs (Ž.K.)
[3] Faculty of Technology, University of Niš, 16000 Leskovac, Serbia; ljiljas76@yahoo.com (L.S.); bojana_d@yahoo.com (B.D.); jelena_stanojevic@yahoo.com (J.S.)
\* Correspondence: zoran.ilic63@gmail.com; Tel.: +381-63-801-4966

**Abstract:** The aim of this study was to determine the antimicrobial activity of essential oils obtained from sweet basil (*Ocimum basilicum* L. cv. 'Genovese') cultivated in the open field under different shading conditions (red, blue, and pearl nets with a shade index of 50% and full sunlight exposure (control plants)), harvested at different times. The antimicrobial activity of basil essential oils (BEOs) obtained from all samples was determined for four microorganisms, while determinations for an additional five microorganisms included samples from non-shaded plants, plants grown under red and pearl nets, and second harvest of plants grown under blue net. Basil essential oil exhibited antimicrobial activity surpassing the activity of relevant commercial antibiotics regardless of growing conditions in the case of *B. cereus*, *K. pneumoniae* and *C. albicans*, while superior antimicrobial activity was exhibited in the case of essential oils from plants grown under blue nets in the case of *S. aureus*, *E. coli* and *P. vulgaris*. The influence of the application of colored shading nets was highly significant ($p < 0.01$) in the cases of all analyzed microorganisms except *C. albicans* and *P. aeruginosa*, while the influence of harvest time was proven in the cases of all microorganisms except *K. pneumoniae*. ANOVA proved that antimicrobial activities are highly dependent on the methods of plant production, shading treatment, and harvest time. Obtained results are discussed in relation to previously determined composition and yield of essential oils from basil grown under shade nets and harvested in different periods.

**Keywords:** *Ocimum basilicum* L.; shading; harvest time; essential oils; antimicrobial activity





## 1. Introduction

Basil essential oils (BEOs) are natural volatile compounds and contain a wide range of chemical constituents produced by plant secondary metabolism [1] and used for applications in various industries such as pharmaceutical [2], cosmetic, perfumery, dental, and oral products [3] and the aromatherapy and food industries [4]. Basil has been applied in traditional medicine with positive effects against fever and cough, flu, asthma, and bronchitis [5]. Basil used in the Balkan diet is described as having radio-protective, anti-inflammatory, anti-stress, anti-diabetic, and anti-pyretic properties [1].

BEOs has been described by their antimicrobial [6,7], anti-bacterial [6,8], anti-fungal [9], and antioxidant properties [8,10]. BEOs are used as natural preservatives for fresh commodity postharvest protection [11] and do not leave any residual effect on fresh produce [12]. BEOs are also used as an important part of bio pesticides [13] as they have repellent, larvicidal, and insecticidal properties [14].

The composition and biological activities of BEOs are highly dependent on the location and environmental conditions [15], chemo types [1], cultivars [16], seasonal variation [17],

different agronomic techniques [18], and different harvesting, drying [19], and processing methods [2].

Net houses have the potential to create an appropriate microclimate that positively affects plants productivity and quality. The reduction of photosynthetically active radiation (PAR) under the colored nets in relation to the open field (full sun light) is different depending on the color of the nets [18]. Light quantity and quality play an important role in the synthesis of EOs. The effect of the light spectrum transmitted by colored shade nets affects the accumulation of secondary metabolites of basil [20] and other plants. Therefore, it is possible to alter the composition of bioactive compound in basil through exposure of plants to a specific light spectrum [21].

Many studies have been published on the antimicrobial activities of EOs from *Lamiaceae*medicinalplants (*Thymus vulgaris* L., *Origanum majorana* L., *Melissa officinalis* L., *Mentha piperita* L., *Origanum vulgare*, and *Ocimum basilicum* L.) [22], as an natural productagainst many different types of microbes, including food-borne pathogens [23]. The inhibition zone is dependent primarily on the medicinal plant and the influence of shading is much less expressed. Our promising findings provide evidence that all EOs from Laminaceae medicinal plants of Serbia exhibit efficacy against resistant pathogenic microorganisms [24]. The results Milenković et al. [24] revealed that EOs from *Thymus vulgaris* L. proved most active against all isolates with an inhibitory zone range between 22 and 56 mm. All the five EOs (*Thymus vulgaris* L., *Origanum majorana* L., *Melissa officinalis* L., *Mentha piperita* L., and *Ocimum basilicum* L.) showed significant anti candida activity. The thyme microbial inhibition zone was largest in the case of *Candida albicans*. Marjoram exhibited the most expressed inhibition in the case of *Pseudomonas aeruginosa*. These two plants exhibited higher inhibition effects in comparison to mint and lemon balm for all other microorganisms included in this investigation [24].

The antimicrobial activity depends on the *Ocimum* cultivar type and also on the developmental stage [25]. The differences in the antibacterial effects of plant extracts may be due to the differences in their phytochemical compositions. Eugenol, the major compound identified in basil oil, has shown activity against protozoa [26] and fungal species, such as *Candida albicans* and *Cryptococcus neoformans* [27]. Methyl chavicol, has been described by its antimicrobial and antioxidant activities [10].

Thyme EO has ahigher antimicrobial potential than basil. Basil used in combination of basil/thyme EOs displayed antagonistic effects against all bacteria, especially against *E. coli* and *S. typhimurium* and obtained a significantly better antimicrobial effect than the commercial antibiotic, gentamicin [28]. Thyme and especially oregano oils showed much larger inhibition zones than other oils and the antibiotic, streptomycin [29].

Having in mind that differences in BEO composition can result in differences in antimicrobial activity, in this study we evaluated the effects of colored shade nets andharvest time on antimicrobial activity of EOs isolated from plants of sweet basil. The cultivar 'Genovese' was grown under the climatic conditions of south Serbia for which the composition and antioxidant activity was already published by Milenković et al. [18] and we compared the BEO antimicrobial activity with commercial antibiotics.

## 2. Material and Methods

### 2.1. Method of Plant Production

The experiment was conducted during 2018–2019 in an experimental garden inthe village Moravac in South Serbia (21°42′ E, 43°30′ N, altitude 159 m). The production and establishment of the basil cultivar 'Genovese'meant sowing seeds in the field at a distance between rows of 40 cm on a plot of only 3 to 6 mm deep on 25 May in raised beds (20 cm high), 1.2 m wide, and 3 m long (3.6 m$^2$ plot size) in 2018. After 6–8 days, the plants began to germinate, and then thinning was performed (at 5 cm distance of plants in the row) to achieve an optimal plant density (50 plants/m$^2$).

Combinations of shade net treatments (pearl, red, and blue color nets, from theIsraeli company Polysack Plastics Industries, with a shade index of 50%) and un-netted (control,

open field) treatments were replicated threetimes in a split-plot design. The shade nets were mounted on a structure placed about 2.0 m above the plants (net house) in themiddle of June until the end of August. Basil plants in the second year (2019) after the establishment of the crop were harvested at the stage of commercial maturity (at thefull-bloom stage). The plants were harvested twice, in early August and early September. Uniform shoots without disease and with leaves without any injuries or defects were selected and dried without the presence of light and ventilation at room temperature (about 25–30 °C) as air-dried plants for analysis.

The growing of basil plants, under colored shade nets and non-shaded as well as the process of production of BEOs by hydrodistillation were performed as described Milenković et al. [18].

## 2.2. Hydrodestillation of BEOs

Essential oil isolation was performed from disintegrated and homogenized plant material (*Basilici herba*) by Clevenger-type hydrodistillation described by Stanojević et al. [30].

## 2.3. Antimicrobial Activity

Antimicrobial activity was determined for nine microorganisms: *E. coli*, *P. vulgaris*, *B. cereus*, *P. aeruginosa*, *B. subtilis*, *K pneumonia*, *S. aureus*, and *L. monocytogenes*and the fungal strain, *C. albicans*, for all treatments except the first harvest of basil grown under blue shading net for which antimicrobial activity was determined for only four microorganisms due to insufficient sample quantity. Microorganisms are obtained from the collection of theFaculty of Technology from Leskovac and Torlak Institute from Belgrade, Serbia [8,30].

The colonies were taken directly from the plate and suspended in 5 mL of sterile 0.85% saline. The turbidity of the initial suspension was adjusted by comparing it with 0.5 McFarland's turbidity [31]. After adjusting to the turbidity of the standard, the bacterium suspension contained about $10^8$ colony-forming units (CFU)/mLand the suspension of fungus contained $10^6$ CFU/mL. Bacterial cell suspensions were inoculated to the nutrient agar plates (Torlak, Belgrade, Serbia) and the fungal suspension to the Sabouraud maltose agar plates (Torlak, Belgrade, Serbia).

Inoculation, incubation, and measurement of inhibition zone diameter were performed by methods as described Stanojević et al. [30]. DMSO was used as negative control. Every measurement was done after three replications and average values were calculated.

## 2.4. Statistical Data Analysis

Statistical analysis of obtained data was performed using software Statistica (version 13, TIBCO software Inc., Palo Alto, CA, USA). ANOVA was used to analyze the significance of the influence of shading conditions on antimicrobial activity with Duncan's multiple range test used for analysis of significance of differences between means. Principal component analysis (PCA) was used for multivariate relationship between antimicrobial activity with the share of the main constituents and the main compound classes of BEOs.

## 3. Results and Discussion

### 3.1. Growing Conditions

Climatic conditions characterizing the growing season, collected from the closest meteorological data recording unit (Kruševac, 21°20′ E, 43°34′ N, altitude 137 m) (http://www.hidmet.gov.rs/latin/meteorologija/klimatologija_godisnjaci.php, accessed on 15 February 2021) indicate that the growing season in which the experiment was conducted was characterized with close to average or above average values for average daily temperatures and under average total sum of insolation in comparison to multiannual mean values. The climatic conditions of southern Serbia are very favorablefor the production of basil throughout the growing season (Table 1).

**Table 1.** Climatic conditions characterizing production season in which investigations were conducted.

| Month | Number of Summer Days (over 25°C in June; over 30 °C for July and August) | Average Temperature Difference from Multiannual Average (°C) | Sum of Insolation Difference from Multiannual Average (h) |
|---|---|---|---|
| June | 27 | 0.8 | −51.4 |
| July | 10 | −0.2 | −72.5 |
| August | 28 | 2.0 | −1.9 |

Source: Republic Hydrometeorological Service of Serbia (http://www.hidmet.gov.rs/, accesed on 15 February 2021).

The smallest reduction wasrecorded withpearl, then withred nets, and the largest PAR reduction is recorded withblue nets. The reduction of PAR also depended on the period during the day. Thus, a greater reduction in PAR was observed in the late afternoon compared to the morning ornoon. Temperature and relative humidity are less exposed to changes under the nets compared to light parameters (Table 2).

**Table 2.** Influence of shading net on growing environment (average day in July).

| Time (h) | PAR [a] Non-Shaded $\mu$mol m$^{-2}$ s$^{-1}$ | Reduction, % Pearl | Red | Blue | Temperature Non-Shaded °C | Reduction, % Pearl | Red Net | Blue | Relative Humidity Non-Shaded % | Reduction, % Pearl | Red | Blue |
|---|---|---|---|---|---|---|---|---|---|---|---|---|
| 6:00 | 182.5 | 31.2 | 39.2 | 46.8 | 16.7 | 0.0 | 0.6 | 1.8 | 74.7 | −4.1 | −5.1 | −5.6 |
| 9:00 | 1325.6 | 46.0 | 48.0 | 54.0 | 24.7 | −0.4 | 0.0 | −1.9 | 71.8 | 0.0 | 0.2 | −0.5 |
| 12:00 | 2242.2 | 48.1 | 50.6 | 56.8 | 31.4 | −2.2 | −3.1 | −1.9 | 47.3 | −2.1 | −2.9 | −3.3 |
| 15:00 | 1684.0 | 51.9 | 51.3 | 59.8 | 31.5 | −3.4 | −1.2 | −0.3 | 48.2 | −1.2 | −1.8 | −2.6 |
| 18:00 | 672.0 | 53.9 | 58.7 | 67.0 | 28.3 | −1.0 | −0.3 | 0.0 | 50.4 | −0.2 | −0.2 | 0.4 |

[a] PAR, photosynthetically active radiation.

### 3.2. Antimicrobial Activity

Antimicrobial activity of sweet basil oil was tested against nine microorganisms, eight bacteria (four Gram-positive and four Gram-negative) and one fungus. BEOs exhibitedantimicrobial effect on all tested microorganisms. Antimicrobial effects are compared against antimicrobial effects of commercial antibiotics. In Table 3 minimum and maximum antimicrobial activities of BEOs, regardless of applied treatments, are provided.

**Table 3.** Antimicrobial activity (inhibition zone, mm) of the basil essential oils in comparison to commercial antibiotics.

| | Microorganism | Inhibition Zone (mm) Essential Oil Antibiotics ColoredNets Pearl | Red | Blue | Control Unshade | A | B | C | N | P |
|---|---|---|---|---|---|---|---|---|---|---|
| Gram-positive bacteria | *Bacillus cereus* | 24.8 ± 1.33 | 21.8 ± 1.17 | 26.5 ± 3.21 | 29.7 ± 1.37 | n.t. | n.t. | n.t. | n.t. | 13.0 |
| | *Bacillus subtilis* | 26.5 ± 7.18 | 27.2 ± 1.60 | 18.3 ± 0.58 | 29.3 ± 0.52 | n.t. | n.t. | 48.0 | n.t. | n.t. |
| | *Listeria monocitogenes* | 24.2 ± 1.47 | 20.2 ± 3.19 | 25.7 ± 1.53 | 14.3 ± 0.52 | n.t. | n.t. | 34.0 | n.t. | n.t. |
| | *Staphylococcus aureus* | 26.0 ± 1.10 | 26.8 ± 2.99 | 38.7 ± 1.15 | 14.3 ± 0.52 | 36.7 | 42.1 | 26.0 | n.t. | 32.3 |
| Gram-negative bacteria | *Proteus vulgaris* | 23.3 ± 2.34 | 24.2 ± 2.14 | 28.0 ± 3.79 | 23.3 ± 0.52 | 13.2 | 22.9 | n.t. | n.t. | n.t. |
| | *Klebsiella pneumoniae* | 34.3 ± 3.44 | 34.0 ± 5.22 | 25.3 ± 1.15 | 21.3 ± 0.52 | n.t. | n.t. | 13.0 | n.t. | 11.7 |
| | *Escherichia coli* | 18.7 ± 1.21 | 26.5 ± 3.56 | 28.3 ± 1.51 | 18.3 ± 0.52 | n.i. | 15.0 | 26.0 | n.t. | n.t. |
| | *Pseudomonas aeruginosa* | 18.8 ± 0.98 | 16.5 ± 1.38 | 18.2 ± 1.60 | 17.7 ± 1.86 | n.t. | n.t. | n.t. | n.t. | 15.7 |
| Fungus | *Candida albicans* | 32.2 ± 5.53 | 29.5 ± 2.07 | 33.7 ± 0.58 | 31.7 ± 1.37 | n.t. | n.t. | n.t. | 17.0 | n.t. |

n.i., no influence; n.t., not treated; A, ampicilin; B, bactrim; C, befalexin; N, nystatin; P, penicillin. All data represent the mean of six replications ± standard deviation.

The strongest inhibition activity of BEOs, surpassing, significantly, the antimicrobial activity of tested antibiotics, regardless of basil harvesting time and application of color shade nets was observed against *B. cereus*, *K. pneumoniae*, and *C. albicans*. Inhibition activity against *B. cereus* was higher in comparison to *Penicillin*, inhibition activity against *K. pneumoniae* in comparison to *Cephalexin* and *Penicillin*, and inhibition activity against *C. albicans* in comparison to *Nystatin*.

Although it is considered that Gram-positive bacteria are more susceptible to growth inhibition by plant EOs than Gram-negative bacteria due to the great complexity of the double-membrane-containing cell envelope in Gram-negative bacteria compared to the single membrane structure of the positive ones [32], in our investigation the most expressed inhibition effect was recorded in the case of one Gram-positive (*B. cereus*) and one Gram-negative (*K. pneumoniae*) bacterium and one fungus (*C. albicans*). EOs can disrupt the permeability barrier of cell membrane structures, with coagulation in the cytoplasm and damage to lipids and proteins with lethal termination [33]. Having in mind that *B. cereus* is among food-borne microorganisms [34], while *K. pneumoniae* and *C. albicans* are among frequent causes of hospital-acquired infections [35], the application of Eos, both as agents for suppression of microbial activity in food and as natural disinfectants in medical facilities, should be further considered. Additionally, having in mind the incidence of *C. albicans* infection in the mouth, genitals, and gastrointestinal tract, BEOs could be used as a valuable natural ingredient in pharmaceutical products for treatment of these infections [36]. Excellent results of BEOs in inhibition of *K. pneumoniae* are of high importance having in mind the fact that this bacterium is highly resistant to antibiotics [37]. *C. albicans* is a fungus that makes normal human microbiota in gastrointestinal and genitourinary tracts. Considering that *C. albicans* is a very persistent microorganism that causes urinary infection, the obtained extract could be used not only in forpreventive but also for therapeutic proposes [38]. Considering that *C. albicans* is a eukaryotic cell, it shares many common biological properties with humans, so commercial antifungal agents used today cause harmful effects and it is necessary to develop new, more effective natural antifungal agents [39].

From the presented comparison of the antimicrobial activity of BEOs against commercial antibiotics it can be noted that in the case of some bacterial inhibition effects surpassing the effects of antibiotics, this is dependent on the growing conditions. Namely, in the case of *S. aureus*, *E. coli*, and *P. vulgaris* only BEOs from basil grown under blue shade net in all conducted trials surpassed the effects of the antibiotics ampicillin, bactrim, cephalexin, and penicillin in the case of *S. aureus*, bactrimandcephalexinin the case of *E. coli*, and Ampicillin and Bactrimin the case of *P. vulgaris*.

The high effect of BEOs from basil produced under blue shade net upon these bacteria suggests the potential use of oil as a natural antimicrobial agent. The presence of enteropathogenic *E. coli* in water and the food products is a reliable indicator of fecal contamination and can cause vomiting and diarrhoea in infants and young children [40]. Having these facts in mind BEOs from basil grown under blue shade net can be used for suppression of *E. coli* in food products and elimination of food borne illnesses caused by this most frequent cause of food borne illnesses. *P. vulgaris* is a Gram-negative bacterium that causes urinary infections [38] pointing at the potential use of BEOs from basil grown under blue net in treatment of this type of infections.

In the cases of *P. aeruginosa* and *L. monocytogenes* as bacteria leading to serious, even fatal outcomes of infection, antimicrobial activity is recorded, but, regardless of production conditions, it did not significantly surpass the antimicrobial activity of effective antibiotics. For *P. aeruginosa*, the antimicrobial activity of BEOs did not differ depending on shading conditions and it was approximately on the level of penicillin. For *L. monocytogenes*, a particularly dangerous pathogen that can survive at the low temperatures offood products kept in the refrigerator [41] shading resulted in higher antimicrobial activity in comparison to non-shaded plants, but the recorded antimicrobial activity regardless of shading net color was still significantly lower in comparison to cephalexin.

Results are presented in Table 4 for four microorganisms for which antimicrobial activity was determined for all shading nets and for both harvest times, and in Table 5 for five microorganisms for which determinations of antimicrobial activity were not conducted for BEOs from basil grown under blue shading net for the first harvest time.

**Table 4.** Influence of harvest time and shading with pearl, red, and blue nets on antimicrobial activity of BEOs.

| | | Inhibition Zone (mm) | | | |
|---|---|---|---|---|---|
| Harvest | Shade Nets | *E. coli* | *P. vulgaris* | *B. cereus* | *P. aeruginosa* |
| First | Pearl | 18.3 [ab] | 23.3 [bc] | 29.7 [d] | 17.7 [ab] |
| | Red | 19.7 [b] | 21.3 [a] | 24.0 [bc] | 18.3 [ab] |
| | Non-shaded | 29.7 [e] | 26.0 [d] | 22.7 [ab] | 15.7 [a] |
| Second | Pearl | 18.3 [ab] | 23.3 [bc] | 29.7 [d] | 17.7 [ab] |
| | Red | 17.7 [a] | 25.3 [d] | 25.7 [c] | 19.3 [b] |
| | Non-shaded | 23.3 [c] | 22.3 [ab] | 21.0 [a] | 17.3 [ab] |
| | | Significance of influence (*p*) | | | |
| Harvest | | 0.000291 | 0.000189 | 0.006826 | 0.041890 |
| Net | | 0.000000 | 0.000000 | 0.000000 | 0.059618 |
| Harvest X Net | | 0.000002 | 0.000000 | 0.000244 | 0.524663 |

Different superscript letters indicate significant differences according to Duncan's multiple range test ($p \leq 0.05$).

**Table 5.** Influence of harvest time and shading with pearl and red nets on antimicrobial activity of BEOs.

| | | Inhibition Zone (mm) | | | |
|---|---|---|---|---|---|
| Harvest | Shade nets | *S. aureus* | *L. monocitogenes* | *B. subtilis* | *K. pneumoniae* | *C. albicans* |
| First | Pearl | 26.3 [b] | 25.0 [d] | 20.0 [a] | 32.0 [b] | 27.3 [a] |
| | Red | 24.3 [b] | 17.3 [b] | 27.7 [bc] | 38.7 [c] | 29.7 [ab] |
| | Non-shaded | 14.3 [a] | 14.3 [a] | 29.3 [c] | 21.3 [a] | 31.7 [b] |
| Second | Pearl | 25.7 [b] | 23.3 [cd] | 33.0 [d] | 36.7 [c] | 37.0 [c] |
| | Red | 29.3 [c] | 23.0 [c] | 26.7 [b] | 29.7 [b] | 29.3 [ab] |
| | Non-shaded | 14.3 [a] | 14.3 [a] | 29.3 [c] | 21.3 [a] | 31.7 [b] |
| | | Significance of influence (*p*) | | | | |
| Harvest | | 0.014985 | 0.011067 | 0.000011 | 0.071116 | 0.004606 |
| Net | | 0.000000 | 0.000000 | 0.003391 | 0.000000 | 0.070349 |
| Harvest XNet | | 0.001229 | 0.000053 | 0.000000 | 0.000033 | 0.000875 |

Different superscript letters indicate significant differences according to Duncan's multiple range test ($p \leq 0.05$).

The presented results show that the effect of application of colored shading nets was highly significant ($p < 0.01$) in the cases of all analyzed microorganisms except the fungus *C. albicans* and bacterium *P. aeruginosa*. The effects of basil harvest time on antimicrobial activity of BEOs was proven with $p < 0.05$ for *S. aureus* and *L. monocytogenes*, and with $p < 0.01$ for all other microorganisms except the bacterium *K. pneumoniae* for which the effects of harvest time were not proven to be significant.

Against *E. coli* superior inhibition was recorded in the case of BEOs from basil grown under red and blue nets for both harvests, against *P. vulgaris*, for BEOs from basil from the second harvest grown under blue net, and against *B. cereus* for BEOs from basil grown under blue net from the second harvest and from the non-shaded plants (Table 4).

Emphasizing the differentiation of antimicrobial activity of BEOs from basil grown under blue net, particularly after prolonged cultivation under shaded conditions characterizing the second harvest points at the potential for application of blue net for enhancement of antimicrobial activity of BEOs.

Superior inhibition against *S. aureus* was exhibited by BEOs from the second harvest plants grown under red net. Against *L. monocytogenes*, the best results were obtained with

BEOs from plants grown under pearl net. BEOs from the second harvest plants grown under pearl netrestrictedthe best growth *B. subtilis*. BEOs from the first harvest plants grown under red net the slowed growth *K. pneumoniae* the most. The great inhibition against *C. albicans* was obtained from BEOs from the second harvest plants grown under pearl net. However, based on observations from Table 4 the investigation of antimicrobial activity of BEOs from basil grown under blue shaded net is promising and should be investigated in future experiments.

### 3.3. Relationship between Antimicrobial Activity and BEO Composition

It would be interesting to analyze the relationships between registered antimicrobials and the composition of BEOs. Thus, the relationships of antimicrobial activity presented in present work with composition of analyzed BEOs presented in our previously published results (Milenković et al. [18]) are considered. The components with the highest share in all analyzed BEOs were linalool (42.3–53.9%) and eugenol (9.7–20.9%), both belonging to oxygen-containing monoterpenes as the class of compounds with the highest content (56.9–70.8%) in examined BEOs [18].

The antimicrobial activity of essential oils dependedon the basil components and their solubility [42]. Higher water solubility compounds like linalool had stronger antimicrobial activity. On the contrary, methyl chavicol had a low antimicrobial activity because of itslower water solubility. The penetration of BEOs in the bacterium or fungus and their antimicrobial activity is strongly related to their solubility in the phospholipids bilayer of cell membranes [43]. Linalool has more effective antibacterial than antifungal activity due to protein denaturation and solvent dehydration [44]. This phenomenon can cause leakage of intracellular components and lead to the death of the bacterial cell [45]. Eugenol-incorporated nanofibers could effectively retard the growth of *Salmonella typhimurium* and *Staphylococcus aureus* and have potential applications as antimicrobial materials in active food packaging [46].

However, attributing the antimicrobial activity of a complex mixture, such as an BEOs to a single constituent is difficult. Major or minor constituents present in the BEOs might be responsible for the antimicrobial activity exhibited [47]. Therefore, possible synergistic or antagonistic properties of the compounds present in the BEOs must be taken into consideration.

The bioactive components of essential oils produce a flux which induces the protons towards the exterior of the microbial cell and causes certain changes and ultimately death of microbial cell [48]. However, according to Bakkali et al. [49] the high volatility of EOs, and their price, as well as their odour and its effect on fresh commodity flavour and taste [50] when used at effective doses, may exceed sensory acceptable levels, are the most common problems encountered in the application of EOs in the food industry.

In order to analyze the relationships between the six major compounds in analyzed BEOs (linalool, 1, 8-cineole, α-terpineol, eugenol, epi-α-cadinol, and α-trans-bergamotene), all belonging to oxygen-containing monoterpenes, principal component analysis (PCA) was performed (Figure 1).

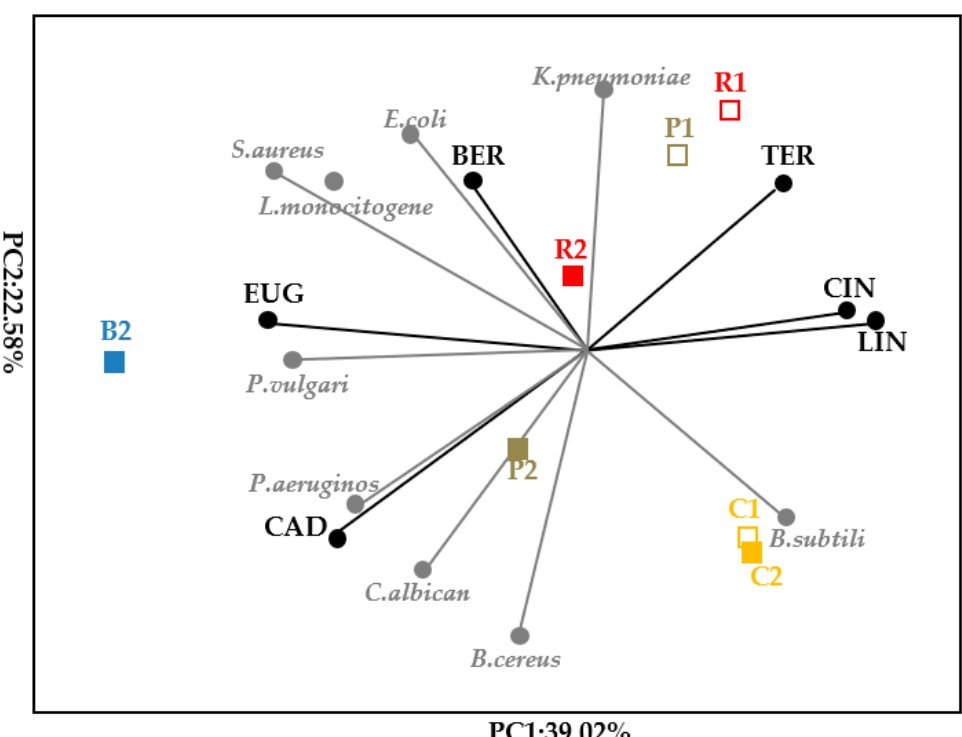

**Figure 1.** The relationships between major compounds in BEOs and antimicrobial activity depend on basil production factors. (Compounds: LIN, linalool; CIN, 1, 8-cineole; TER, α-terpineol; EUG, eugenol; CAD, epi-α-cadinol; BER, α-trans-bergamotene. Shading nets and harvests: C1 and C2, non-shaded first and second harvest; R1 and R2, red net first and second harvest; P1 and P2, pearl net first and second harvest; B2, blue net second harvest).

The differentiation based on the first principal component (PC1), explaining 39.02% of the total variability, showed that BEOs from shadedplats from the second harvest, had increased contents of eugenol and decreased content of linalool and 1.8-cineole. Higher eugenol content seemed to be associated withhigher antimicrobial activity against *P. vulgaris* related primarily to BEOs grown under blue net.

The second principal component (PC2) which explains 22.58% of the variability can probably be attributed to differentiation of BEOs from non-shaded and shaded plants, with higher antimicrobial activity against *B. subtilis* exhibited by BEOs from non-shaded plants.

PCA points at the probable relationship between the content of particular compounds in BEOs and the antimicrobial activity. It seems that a higher share of eugenol can be related to increased inhibition of *P. vulgaris*, *E. coli*, *S. aureus, L. monocitoges, P. aeruginosa, and C. albicans*. A higher share of α-trans-bergamotene seems to be related to antimicrobial activity against *E. coli*, *S. aureus*, and *L. monocitogenes*, while higher epi-α-cadinol share seems to be related toantimicrobial activity against *P. aeruginosa*, *P. vulgaris*, *B. sereus*, and *C. albicans*. No direct relationship between an increased share of linalool, 1, 8-cineole, or α-terpineol andantimicrobial activity of BEOs againstspecific microorganismscould be noted from PCA.

Principal component analysis was performed also in order to analyze the relationship between the seven classes of compounds found in analyzed BEOs (oxygen-containing monoterpenes, monoterpene hydrocarbons, sesquiterpene hydrocarbons, oxygen-containing sesquiterpenes, aliphatic alcohols, aliphatic esters, and aromatic compounds) presented in Figure 2.

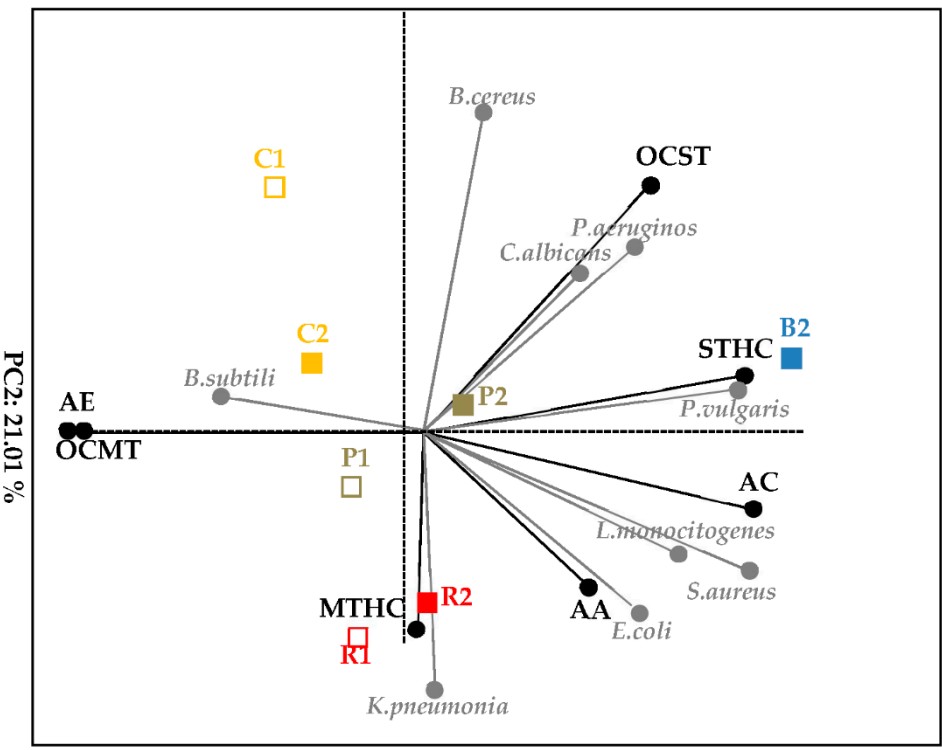

**Figure 2.** The relationships between classes of compounds in BEOs and antimicrobial activity depend on basil production factors. (Classes of compounds: OCMT, oxygen-containing monoterpenes; MTHC, monoterpene hydrocarbons; STHC, sesquiterpene hydrocarbons; OCST, oxygen-containing sesquiterpenes; AE, aliphatic esters; AA, aliphatic alcohols; and AC, aromatic compounds. Shading nets and harvests: C1 and C2, non-shaded first and second harvest; R1 and R2, red net first and second harvest; P1 and P2, pearl net first and second harvest; and B2, blue net second harvest).

The second principal component (PC2) which explains 21.01% of the variability points at differentiation of BEOs from basil grown under blue net in comparison to all other BEOs. This differentiation can be attributed to a higher content of sesquiterpene hydrocarbons and lower content of aliphatic esters and oxygen-containing monoterpenesin BEO from basil grown under blue net which can be associated withhigher antimicrobial activity against *P. vulgaris* and lower antimicrobial activity against *B. subtilis*.

The first principal component (PC1) which explains 44.14% of variability points at differentiation of BEOs from basil grown under red net in comparison to all other BEOs based on higher content of monoterpene hydrocarbons which can be associated with higher antimicrobial activity against *K. pneumoniae*.

PCA indicates the same trends, pointing at the probable relationship that can be noticed in the case shared antimicrobial activity against *L. monocytogenes*, *S. aureus*, and *E. coli* of aromatic components and aliphaticalcohols and the oxygen-containing sesquiterpenes' shared antimicrobial activity against *P. aeruginosa* and *C. albicans*.

BEOs revealed antibacterial properties, but the degree of bacterial growth inhibition induced by the basil plant was shown to be related to bacterial strain and herbal characteristics (light manipulation by color shade nets and harvest time optimization).

The inhibition activity of BEOs significantly surpassedthe antimicrobial activity of relevant antibiotics, regardless of harvesting time and application of color shade nets in cases of *B. cereus*, *K. pneumoniae*, and *C. albicans* while in the case of *S. aureus*, *E. coli*, and *P. vulgaris* only BEOs from basil grown under blue shade net surpassed the effects of antibiotics.

BEOs from shaded plats from the second harvest, primarily BEOs grown under blue net, led to increased contents of eugenol, which can be associated withhigher antimicrobial

activity against *P. vulgaris* and decreased contents of linalool and 1, 8-cineole. Higher antimicrobial activity against *B. subtilis* was exhibited by BEOs from non-shaded plants. BEOs from basil grown under red net were characterized with higher monoterpene hydrocarbons content which can be associated with higher antimicrobial activity against *K. pneumoniae*, while BEOs from basil grown under blue net were differentiated by having a higher content of sesquiterpene hydrocarbons which can be associated withhigher antimicrobial activity against *P. vulgaris*.

**4. Conclusions**

Growing basil under blue net, particularly in the case of late harvest, seems to be promising for enhancement of the antimicrobial activity of BEOs. Photo-selective shading provided by the blue net could be, based on presented results, incorporated into the protected cultivation practices currently used for cultivation of basil for essential oil production. BEOs from plants grown under blue nets are characterized byhigher eugenol content and show superior antimicrobial activity against *S. aureus*, *E. coli*, and *P. vulgaris*. The promising antimicrobial activity of BEOs has encouraged researchers to investigate and to use them with nanomaterials, or to use BEOs in combination with EOs from other plants, as potential antimicrobial agents as an alternative to chemicals and conventional antibiotics (ampicilin, bactrim, cefalexin, nystatin, and penicilin).

**Author Contributions:** Z.S.I. and J.M., (head of the research group) planned the research, analyzed the data, and wrote the manuscript; L.M. and L.Š. conducted the experiments in the field; J.S., B.D., and N.T. performed analyses of physical properties and chemical composition in the laboratory; L.S. and Ž.K. performed statistical analysis of obtained results. All authors have read and agreed to the published version of the manuscript.

**Funding:** This study, which was part of the projects TR-31027 and TR-34012 (Program for financing scientific research work, number 451-03-9/2021-14/200133 and 451-03-9/2021-14/200222) was financially supported by the Ministry of Education Science and Technological Development of the Republic of Serbia.

**Institutional Review Board Statement:** Not applicable.

**Informed Consent Statement:** Not applicable.

**Data Availability Statement:** Publicly available meteorological datasets from Republic Hydrometeorological service of Serbia, which are presented and analyzed in this study. This data (in Serbian language) can be found here: http://www.hidmet.gov.rs/latin/meteorologija/klimatologija_godisnjaci.php, accessed on 15 February 2021.

**Conflicts of Interest:** The authors declare no conflict of interest.

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
