# Peer review of "Efficiency of Basil Essential Oil Antimicrobial Agents under Different Shading Treatments and Harvest Times"

_agronomy, doi:10.3390/agronomy11081574_

Round 1

Reviewer 1 Report

The paper reports some interesting results that justify publication since the topic clearly fit to the journal scoop. Please consider the following comment to revise the manuscript:

  • Authors are using both GB and US English please revise the manuscript using uniform style
  • What is the significance of red text in table 3 and table 4
  • In table 2: the reduction of different parameters is confusing!! When the reduction is negative (Reduction, %) it means there is an increasing temperature!!?
  • Based on the PCA analysis what is the outcome from figure 1 and 2. Are there any structuration of relationship between shading condition, major compounds, and the antimicrobial activity!!
  • The conclusion does not clearly show the importance of this study and the originality of the obtained results. Please state clearly what make this work original and the topic worthy of investigation.

Author Response

Reviewer 1

Comments and Suggestions for Authors

The paper reports some interesting results that justify publication since the topic clearly fit to the journal scoop. Please consider the following comment to revise the manuscript:

  • Authors are using both GB and US English please revise the manuscript using uniform style

Yes we changes with GB …will be uniform ….colour

  • What is the significance of red text in table 3 and table 4.

Statistically significant represent comparison of BEOs from shading with BEOs from unshading plants

  • In table 2: the reduction of different parameters is confusing!! When the reduction is negative (Reduction, %) it means there is an increasing temperature!!?

Reduction can’t be positive, in comparison with temperature from the open field condition.  Temperature under different nets is lower (reduction) than temperature from un-shading-control conditions!!!!!!!

  • Based on the PCA analysis what is the outcome from figure 1 and 2. Are there any structuration of relationship between shading condition, major compounds, and the antimicrobial activity!!

In figure 1 present relation among major compound in BEOs and antimicrobial activity in dependence of basil production factors (shading colour nets or non-shaded plants)

In figure 2. We show relations among classes of compounds in BEOs and antimicrobial activity in dependence of basil production factors (shading colour nets or non-shaded plants)

  • The conclusion does not clearly show the importance of this study and the originality of the obtained results. Please state clearly what make this work original and the topic worthy of investigation.

Yes we make new formulation of conclusion based on the results achieved and clearly show originality of our study with application of BEOs as alternative of conventional antibiotics.

Submission Date

18 July 2021

Date of this review

27 Jul 2021 10:26:58

29 Jul 2021 17:03:44

Reviewer 2 Report

The study, given the current problem of reducing synthetic pesticides and the increase in demand for natural products, is of considerable interest.

The manuscript needs a thorough revision and it will be possible to positively evaluate it after the required changes. The manuscript must be prepared using the template file Agronomy Microsoft World.

Introduction

It should be enriched with further studies on the use of EO to combat microorganisms, extracted from other plants and belonging to the Lamiaceae family.

The comparison of basil EOs with commercial antibiotics is missing from the objectives.

Material and methods

Line 87 – 102: the paragraph should only report the methodology and tools used to conduct the experiment. This paragraph should indicate the type of instruments and the location of the climatic device, as well as the description of the types of networks used. The data collected by the climatic station and the influence of the networks on the cultivation environment should be reported in the results.

Results

The paragraph does not provide a concise and precise description of the experimental results. The data in the tables are not discussed in the paragraph.

Table 3: improve data visibility.

Line 150: authors should report SD in table.

Table 4-5: missing units of measure.

Line 239: insert reference.

Author Response

Reviewer 2

Open Review

Comments and Suggestions for Authors

The study, given the current problem of reducing synthetic pesticides and the increase in demand for natural products, is of considerable interest.

The manuscript needs a thorough revision and it will be possible to positively evaluate it after the required changes. The manuscript must be prepared using the template file Agronomy Microsoft World.

Introduction

It should be enriched with further studies on the use of EO to combat microorganisms, extracted from other plants and belonging to the Lamiaceae family.

Yes we add some new references on the use of EOs extracted from other plants and belonging to the Lamiaceae family special from our experiments published from this year!!!!!!!

Many studies have been published on the antimicrobial activities of EOs from Lamiaceae medicinal plants (Thymus vulgaris L., Origanum majorana L.Melissa officinalis L., Mentha piperita L.,  Origanum vulgare and Ocimum basilicum L.) [22], as an natural product, against many different types of microbes, including food-borne pathogens [23]. Inhibition zone is dependent primarily on medicinal plant and the influence of shading is much less expressed. Our promising findings provide evidence that all EOs from Laminaceae medicinal plants of Serbia exhibits efficacy against resistant pathogenic microorganisms[24]. The results Milenković et al.[24] revealed that EOs from Thymus vulgaris L., proved most active against all isolates with inhibitory zone range between 22 and 56 mm. All the five EOs (Thymus vulgaris L., Origanum majorana L.Melissa officinalis L., Mentha piperita L. and Ocimum basilicum L.) showed significant anti candida activity. Microbial inhibition zone is in the case of thyme largest in the case of Candida albicans. Marjoram exhibits the most expressed inhibition in the case of Pseudomonas aeruginosa. These two plants exhibit higher inhibition effects in comparison to mint and lemon balm for all other microorganisms included in this investigation [24].

  1. Avetisyan, A.; Markosian, A.; Petrosyan, M.; Sahakyan, N.; Babayan, A.; Aloyan, S.; Trchounian, A. Chemical composition and some biological activities of the essential oils from basil Ocimum different cultivars. BMC Complem. Alter Med .2017, 17, 60.
  2. Kozłowska, M.; Laudy, A.E.; Przybył, J.; Ziarno, M.; Majewska, E. Chemical composition and antibacterial activity of some medicinal plants from Lamiaceae family. Acta Pol Pharm. 2015, 72(4), 757-67. PMID: 26647633
  3. 24. Milenković, L.; Ilić, S.Z.; Šunić, Lj.; Mastilović, J.; Kevrešan, Ž.; Cvetković, D.; Stanojević, L.; Danilović, B.; Stanojević, J. Antimicrobial potential of essential oils from Lamiaceae medicinal plants of Serbia. 2021, In press…..

The comparison of basil EOs with commercial antibiotics is missing from the objectives.

Yes we add comparison of basil EOs with commercial antibiotics with two new references

  1. 28. Semeniuc, A.; Rodica C.; Pop.; Rotar A M.. Antibacterial activity and interactions of plant essential oil combinations against Gram-positive and Gram-negative bacteria. J. Food Drug Anal, 2017, 25(2),403-408.
  2. Soković, M.; Glamočlija, J.; Marin, P.D.; Brkić, D.; van Griensven, L.J.L.D. Antibacterial effects of the essential oils of commonly consumed medicinal herbs using an in vitro model. Molecules. 2010, 15, 7532-7546.

Material and methods

Line 87 – 102: the paragraph should only report the methodology and tools used to conduct the experiment. This paragraph should indicate the type of instruments and the location of the climatic device, as well as the description of the types of networks used. The data collected by the climatic station and the influence of the networks on the cultivation environment should be reported in the results.

Yes, this paragraph we remove from Material and Methods to the - Results !!!!!!

Results

The paragraph does not provide a concise and precise description of the experimental results. The data in the tables are not discussed in the paragraph.                                                                                                                              Table 3: improve data visibility.

We are improving data visibility.

Line 150: authors should report SD in table.

We are add standard deviations

All data represent the mean of six replications ± standard deviation

Table 4-5: missing units of measure.

Yes, we add ….Inhibition  zone (mm)

Line 239: insert reference.

We add reference…

  1. Chouhan, S.;  Sharma, K.; Guleria, S. Antimicrobial activity of some essential oils-Present status and future perspectives. Medicines(Basel) 2017, 4(3),58.

Submission Date

18 July 2021

Date of this review

29 Jul 2021 17:03:44

Round 2

Reviewer 1 Report

the current version of the paper seems revised according to the comments. The revised manuscript may be considered for publication. I have no further comments.

Reviewer 2 Report

the authors have made the required changes, the work is improved,
I congratulate you on accepting my advice